# Pentraxin 3 Inhibits the Angiogenic Potential of Multiple Myeloma Cells

**DOI:** 10.3390/cancers13092255

**Published:** 2021-05-08

**Authors:** Roberto Ronca, Sara Taranto, Michela Corsini, Chiara Tobia, Cosetta Ravelli, Sara Rezzola, Mirella Belleri, Floriana De Cillis, Annamaria Cattaneo, Marco Presta, Arianna Giacomini

**Affiliations:** 1Department of Molecular and Translational Medicine, University of Brescia, 25123 Brescia, Italy; sara.taranto@unibs.it (S.T.); michela.corsini@unibs.it (M.C.); chiara.tobia@unibs.it (C.T.); cosetta.ravelli@unibs.it (C.R.); sara.rezzola@unibs.it (S.R.); mirella.belleri@unibs.it (M.B.); marco.presta@unibs.it (M.P.); 2Biological Psychiatry Unit, IRCCS Istituto Centro San Giovanni di Dio Fatebenefratelli, 25125 Brescia, Italy; fdecillis@fatebenefratelli.eu (F.D.C.); acattaneo@fatebenefratelli.eu (A.C.); 3Department of Pharmacological and Biomolecular Sciences, University of Milan, 20122 Milan, Italy

**Keywords:** multiple myeloma, long pentraxin 3, FGF/FGFR system, angiogenesis

## Abstract

**Simple Summary:**

Bone marrow (BM) angiogenesis represents a key aspect in the progression of multiple myeloma (MM) and is strictly linked to the balance between pro-angiogenic and anti-angiogenic players produced by both neoplastic and stromal components. It has been shown that Fibroblast Growth Factors (FGFs) play a pivotal role in the angiogenic switch occurring during MM progression. Accordingly, the natural FGF antagonist Long Pentraxin 3 (PTX3) is able to reduce the activation of BM stromal components induced by FGFs. This work explores, for the first time, the anti-angiogenic role of PTX3 produced by MM cells demonstrating that the inducible expression of PTX3 is able to impair MM neovascularization, the onset of a proficient BM vascular niche and, ultimately, to impair tumor growth and dissemination.

**Abstract:**

During multiple myeloma (MM) progression the activation of the angiogenic process represents a key step for the formation of the vascular niche, where different stromal components and neoplastic cells collaborate and foster tumor growth. Among the different pro-angiogenic players, Fibroblast Growth Factor 2 (FGF2) plays a pivotal role in BM vascularization occurring during MM progression. Long Pentraxin 3 (PTX3), a natural FGF antagonist, is able to reduce the activation of stromal components promoted by FGF2 in various in vitro models. An increased FGF/PTX3 ratio has also been found to occur during MM evolution, suggesting that restoring the “physiological” FGF/PTX3 ratio in plasma cells and BM stromal cells (BMSCs) might impact MM. In this work, taking advantage of PTX3-inducible human MM models, we show that PTX3 produced by tumor cells is able to restore a balanced FGF/PTX3 ratio sufficient to prevent the activation of the FGF/FGFR system in endothelial cells and to reduce the angiogenic capacity of MM cells in different in vivo models. As a result of this anti-angiogenic activity, PTX3 overexpression causes a significant reduction of the tumor burden in both subcutaneously grafted and systemic MM models. These data pave the way for the exploitation of PTX3-derived anti-angiogenic approaches in MM.

## 1. Introduction

Multiple myeloma (MM) represents a life-threatening hematological disorder, being the second most common blood cancer diagnosis, with over 12,000 deaths estimated per year and a 5-year survival rate around 54% (www.cancer.net, accessed on 7 May 2021). Notwithstanding the introduction of novel therapeutic approaches that have led to steadily increased survival rates over the last decade, a complete eradication has not been obtained so far, and MM remains an incurable disease [1,2].

In the complex neoplastic MM microenvironment, tumor growth and resistance are fostered not only by plasma cells themselves, but also by bone marrow (BM) stromal cells (BMSCs), including endothelial cells (ECs). Indeed, angiogenesis represents a key feature of MM progression, driving the transition from the avascular state of monoclonal gammopathies of undetermined significance (MGUS) to the widely vascularized condition of active MM [3,4]. Accordingly, BM microvascular density represents a significant prognostic factor for progression free and overall survival in MM patients [5,6].

Among the pro-angiogenic factors, Fibroblast Growth Factor 2 (FGF2) has been shown to play a relevant role in different tumor types, including MM [7,8,9], its blockade resulting in significant anti-tumor and anti-angiogenic activities [10,11]. Accordingly, the pattern recognition receptor Long Pentraxin 3 (PTX3), a secreted/stromal component of innate immunity able to bind and inactivate various members of the FGF family, including FGF2, has revealed potent anti-angiogenic and anti-tumor properties in different FGF-dependent tumors [12,13,14,15,16,17]. In a translational perspective, a PTX3-derived FGF trap molecule has been proposed for the treatment of these tumors, including MM [11,18,19,20].

PTX3 and FGF2 are actively produced by BM plasma cells, ECs, and fibroblasts in normal, MGUS, and MM settings. However, the ratio between PTX3 and FGF2 decreases during the transition from MGUS to MM, leading to more abundant levels of FGF2 in these cells, representing a brake release mechanism to promote an angiogenic switch in MM [21]. Accordingly, in vitro observations have shown that treatment with recombinant PTX3 impaired FGF-mediated viability, chemotaxis, and migration of ECs and fibroblasts isolated from the BM of MM patients, as well as plasma cell adhesion to these cells [21]. These data suggest that restoring physiological PTX3/FGF ratio in plasma cells and BMSCs might impact MM. 

To date, no data are available about the effect exerted in vivo by PTX3 of plasma cell origin on the growth and vascularization of MM. Here, taking advantage from PTX3-inducible human MM cell models, we demonstrate that PTX3 upregulation in plasma cells strongly impact MM growth and dissemination, mainly impairing FGF-mediated angiogenesis. These data add new hints regarding the role of PTX3 in MM and pave the way for the exploitation of PTX3-derived anti-angiogenic approaches in MM.

## 2. Materials and Methods

### 2.1. Cell Cultures and Reagents

KMS-11 cells were obtained from the Japanese Collection of Research Bioresources (JCRB, Osaka, Japan) cell bank; GFP/Luciferase–expressing MM.1S cells were from Dr. Ghobrial (Dana-Farber Cancer Institute, Boston, MA, USA). All cell lines were maintained at low passage in RPMI1640 medium supplemented with 10% heat-inactivated FBS and 2.0 mM glutamine, tested regularly for *Mycoplasma* negativity, and authenticated by PowerPlex Fusion System (Promega, Madison, WI, USA). KMS-11 PTX3/Mock and MM.1S PTX3/Mock cells were obtained by lentiviral transduction using the pLVX-TetOne-Puro vector (Clontech, Mountain View, CA, USA), either harboring or not harboring the human PTX3 coding sequence. Cells were selected adding 1 µg/mL of puromycin to the cell culture medium. Doxycycline (resuspended at 1.0 mg/mL) was purchased from Merck.

### 2.2. Western Blot Analysis

Cells were washed in cold PBS and homogenized in NP-40 lysis buffer (1% NP-40, 20 mM Tris–HCl pH 8, 137 mM NaCl, 10% glycerol, 2 mM EDTA, 1 mM sodium orthovanadate, 10 μg/mL aprotinin, 10 μg/mL leupeptin). Protein concentration in the supernatants was determined using the Bradford protein assay (Bio-Rad Laboratories, Hercules, CA, USA). Expression of PTX3 was detected using anti-PTX3 rabbit polyclonal antibody (from B. Bottazzi, Humanitas Clinical Institute, Rozzano, Italy). GAPDH antibody (Santa Cruz Biotechnology, Santa Cruz, CA, USA) and red Ponceau staining were used as loading controls for cell lysates and conditioned medium, respectively. Chemiluminescent signal was acquired by ChemiDoc™ Imaging System (Bio-Rad Laboratories, Hercules, CA, USA). 

### 2.3. RT-qPCR

Total RNA was extracted using TRIzol Reagent (Invitrogen, Carlsbad, CA, USA) according to the manufacturer′s instructions. Two µg of total RNA were retro-transcribed with MMLV reverse transcriptase (Invitrogen) using random hexaprimers. Then, cDNA was analyzed by quantitative PCR using the following primers: human *PTX3*, 5′-GTGCTCTCTGGTCTGCAGTG-3′ (forward) and 5′-TCGTCCGTGGCTTGCAGCAG-3′ (reverse); human *CCDN1,* 5′-AATGACCCCGCACGATTTC-3′ (forward) and 5′-CATGGAGGGCGGATTGGAA-3′ (reverse); human *GAPDH,* 5′-TGCCATCACTGCCACCCAGA-3′ (forward) and 5-CGCGGCCATCACGCCACAG-3′ (reverse).

### 2.4. Gene Expression Profiling (GEP)

GEP was performed on KMS-11 PTX3 cells either treated or untreated with DOXA (200 ng/mL) for 96 h. A cut-off of *p*-value < 0.01 (FDR corrected) and Log_2_ fold change ±2 was applied to select differentially expressed genes. Total RNA was extracted using TRIzol Reagent according to the manufacturer’s instructions (Invitrogen). RNA integrity and the purity of the treated cells were assessed using a Bioanalyzer (Agilent Technologies, Santa Clara, CA, USA). Hybridization to an Illumina Microarray (Illumina) was performed. Robust spline normalization and L2T were performed in R software, using the Lumi package from Bioconductor open-source software (http://www.bioconductor.org/, accessed on 7 May 2021). Normalized data were imported into Partek Genomic Suite 6.6 software (Partek). After quality controls, an analysis of variance (ANOVA) test was performed to assess the effects of PTX3 on pro-/anti- angiogenic gene expression, comparing KMS-11 PTX3 cells that received DOXA vs KMS-11 PTX3 cells that did not receive DOXA. 

### 2.5. Cytofluorimetric Analyses

Cytofluorimetric analyses were performed using the MACSQuant^®^ Analyzer (Miltenyi Biotec, Bergisch Gladbach, Germany). Propidium iodide staining (Immunostep, Salamanca, Spain) was used to detect PI negative viable cells and viable cell counts were obtained by the counting function of the MACSQuant^®^ Analyzer. 

### 2.6. MM/HUVE Cells Co-Cultures

KMS-11 PTX3 and MM.1S PTX3 cells were co-cultured with HUVE cells at 15:1 MM/HUVEC ratio in presence or absence of Doxycycline (DOXA) 200 μg/mL. After 48 h of co-culture, MM cells were removed without detaching endothelial cells and HUVEC viable cell counting was performed by cytofluorimetric analysis. 

### 2.7. In Vitro Immunofluorescence Analysis

HUVE cells were seeded in 2.5% FBS in Ibidi^®^ μ-Slide 8 wells (Ibidi, Martinsried, Germany) at a density of 30,000 cells/cm^2^, and co-cultured or not with KMS-11 PTX3 or MM.1S PTX3 in absence or presence of DOXA 200 ng/mL. After 24 h MM cells were removed and HUVE cells were washed twice in PBS, fixed in cold acetone for 5 min, and permeabilized with 0.2% Triton-X100 in PBS for 2 min at RT. After washing in PBS, cells were blocked for 10 min at RT in 1% BSA and then incubated with rabbit anti-pFGFR1 (Tyr766h, Santa Cruz Biotechnology, Dallas, TX, USA) antibody for 1 h at RT. Cells were then washed in PBS and incubated with AlexaFluor 594-conjugated anti-rabbit antibody (Invitrogen) and DAPI for 30 min at RT. Finally, cells were examined under a Zeiss Fluorescence Axiovert 200M microscope (Carl Zeiss, Milan, Italy). 

### 2.8. Chick Embryo Chorioallantoic Membrane (CAM) Assay

Alginate beads (5 μL) containing vehicle or KMS-11 PTX3 cells (40 × 10^4^ cells/implant) with or without DOXA (200 μg/mL) were placed onto the top of chicken embryo CAMs at day 11 of incubation. After 72 h, newly formed blood vessels were quantified as described [22,23,24]. 

### 2.9. Zebrafish Embryo Model

Zebrafish experiments were performed as approved by the local animal ethics committee (OPBA, Organismo Preposto al Benessere degli Animali, Università degli Studi di Brescia, Italy). Embryos from the Tg (fli1:egfp) strain of the zebrafish, *Danio rerio*, were collected, staged, and raised at 28.5 °C, according to standard experimental conditions. Embryos at 48 h post-fertilization (hpf) were anesthetized using 0.04 mg/mL of tricaine (Sigma-Aldrich, St. Louis, MO, USA), and placed onto an agarose gel plate meld for tumor cell microinjection. MM.1S PTX3 cells cultured for 48 h in the presence or absence of DOXA (200 ng/mL) were washed and transferred using a micro-loader tip (Eppendorf, Milan, Italy) into a borosilicate glass needle (outer diameter/inner diameter: 1.2/0.68 mm) connected to a FemtoJet microinjector and InjectMan NI 2 Micromanipulator (Eppendorf). Finally, tumor cells were injected into the perivitelline space of embryos under a stereo-dissecting microscope (Leica, MZ75). Twenty-four hours after tumor injection, the angiogenic response was analyzed by quantifying the cumulative length of sprouts originating from the subintestinal vein vessels after phosphatase staining. 

### 2.10. Subcutaneous Human Xenografts

Experiments were performed according to the Italian laws (D.L. 116/92 and following additions) that enforce the EU 86/109 Directive and were approved by the local animal ethics committee (OPBA, Organismo Preposto al Benessere degli Animali, Università degli Studi di Brescia, Italy). Six- to eight-week old female NOD/SCID mice (Envigo, Udine, Italy) were injected subcutaneously (s.c.) with KMS-11 PTX3 (5 × 10^6^ cells/mouse) in 200 µL of PBS. The day after tumor implantation, mice were randomly assigned to receive DOXA (1 mg/mL) in the drinking water. Tumor volumes were measured with caliper and calculated according to the formula V = (D × d^2^)/2, where D and d are the major and minor perpendicular tumor diameters, respectively. At the end of the experimental procedure, mice were injected intravenously with sulfobiotin as previously described [25] in order to label the whole functional vascular network. Finally, tumor nodules were excised and processed for histological analysis.

### 2.11. Systemic Human Xenograft

Experiments were performed according to the Italian laws (D.L. 116/92 and following additions) that enforce the EU 86/109 Directive and were approved by the local animal ethics committee (OPBA, Organismo Preposto al Benessere degli Animali, Università degli Studi di Brescia, Italy). Six- to eight-week old female SCID Beige mice (Envigo) were injected intravenously (i.v) with GFP/Luciferase expressing MM.1S PTX3 cells (2 × 10^6^ cells/mouse) in 100 µL of PBS. The day after tumor cell injection, mice were randomly assigned to receive DOXA (1 mg/mL) in the drinking water. Tumor dissemination was assessed by bioluminescence imaging analysis performed at 3, 4, and 5 weeks after tumor cell injection. 

### 2.12. Histological Analyses

Tumor samples and femurs were either embedded in OCT compound and immediately frozen or fixed in formalin and embedded in paraffin, respectively. 

For immunofluorescence analysis, tumor cryostat sections (5 μm thick) were air dried and fixed with cold acetone (5 min at 4 °C). After blocking with 1% BSA in PBS for 10 min, samples were incubated for 1 h at room temperature with primary antibodies [rabbit anti-PTX3 (from B. Bottazzi, Humanitas Clinical Institute, Rozzano, Italy), rabbit anti-pFGFR1 (Santa Cruz Biotechnology), rat anti-mouse Ki67 (Dako), or rabbit anti-human phospho Histone H3 (Merck)]. After washing with PBS containing 0.05% Tween 20, samples were incubated for 30 min with the appropriate Alexa Fluor 594-conjugated secondary antibody (Invitrogen). In vivo biotinylated endothelial cells were detected by incubating tumor sections with 488-conjugated streptavidin (Invitrogen). Finally, after mounting in a drop of anti-bleaching mounting medium containing DAPI (Vectashield, Vector Laboratories, Burlingame, CA, USA), samples were examined under a Zeiss Fluorescence Axiovert 200M (Carl Zeiss) microscope. 

Formalin-fixed, paraffin-embedded femur samples were sectioned at a thickness of 3 µm, dewaxed, hydrated, and stained with hematoxylin and eosin (H&E) or processed for immunohistochemistry with mouse anti-human CD38 (Novocastra, Wetzlar, Germany), rabbit anti-human PTX3 (from B. Bottazzi, Humanitas Clinical Institute, Rozzano, Italy), or rabbit anti-mouse KDR (Cell Signaling Technology, Danvers, MA, USA) antibodies. A positive signal was revealed by 3,3′-diaminibenzidine (Roche) or Vector blue substrate (Vector Laboratories, Burlingame, CA, USA) stainings. Sections were finally counterstained with Carazzi′s hematoxylin before analysis by light microscopy. Images were acquired with the automatic high-resolution scanner Aperio System (Leica Biosystems, Wetzlar, Germany). Image analysis was carried out using the open-source ImageJ software.

### 2.13. Two-Photon Microscopy

After euthanasia, mice were transcardially perfused with 0.01 M phosphate-buffered saline (PBS) (Sigma-Aldrich, Milan, Italy) followed by 4% paraformaldehyde (PFA) (VWR). After specimen preparation as previously described [26], femurs were incubated for 48 h at 4 °C with rabbit anti-mouse KDR (Cell Signaling Technology) and mouse anti-human CD38 (Novocastra) followed by 4 h incubation with AlexaFluor 594 and AlexaFluor488-conjugated secondary antibodies. Then, samples were stored in PBS at 4 °C. For imaging, bones were held in 1% low melting agarose. Two-photon imaging was performed on a Zeiss LSM880 equipped with an EC Plan-Neofluar 20×/0.50 controlled by Zen Black 2 (Zeiss GmbH, Oberkochen, Germany). 

### 2.14. Statistical Analyses 

Statistical analyses were performed using Prism 8 (GraphPad Software). Student′s *t-* test for unpaired data (2-tailed) was used to test the probability of significant differences between two groups of samples. For more than two groups of samples, data were analyzed with a 1-way analysis of variance and corrected by the Bonferroni multiple comparison test. Tumor volume data were analyzed with a 2-way analysis of variance and corrected by the Bonferroni test. Differences were considered significant when *p* < 0.05. 

## 3. Results

### 3.1. PTX3 Produced by MM Cells Hampers the Proliferation of Endothelial Cells

To assess the role of PTX3 expressed and released by MM cells, we generated two human MM cell lines with doxycycline-inducible expression of PTX3 (KMS-11 PTX3 and MM.1S PTX3 cells). As shown in Figure 1A, KMS-11 PTX3 and MM.1S PTX3 cells express and secrete high levels of PTX3 already 48 h after treatment with 200 ng/mL of doxycycline (DOXA) when compared to untreated (-DOXA) or control (Mock) cells. In keeping with the capacity of MM cells to stimulate ECs by direct interactions and production of pro-angiogenic factors, including FGF2 [2], human umbilical vein ECs (HUVECs) co-cultured with MM cells showed an increased rate of survival and proliferation as well as elevated levels of FGFR1 phosphorylation, when compared to HUVEC monocultures (Figure 1B,C). Notably, PTX3 released by MM cells upon DOXA induction significantly reduced HUVEC proliferation and FGFR1 phosphorylation to the basal levels observed in HUVEC monocultures (Figure 1B,C). 

It must be pointed out that, in keeping with previous observations [21], PTX3 upregulation following DOXA induction did not directly affect the survival and proliferation of MM cells (Appendix A), thus ruling out the possibility that the inhibitory effects observed in ECs were due to a reduced survival/proliferation of MM cells. Gene expression profiling performed on KMS-11 PTX3 cells also did not show any significant modulation of the expression of *FGF2* and other pro-/anti-angiogenic genes upon PTX3 induction (Appendix A), indicating that the inhibition of EC proliferation is due to the FGF-trap activity of PTX3 rather than to a modulation of other pro- or anti-angiogenic factors caused by PTX3 overexpression.

Together, these data are in keeping with the capacity of FGF2 to act as a paracrine survival/proliferation factor for ECs and with the potent anti-angiogenic activity of PTX3 consequent to its FGF trap activity that results in the inhibition of the FGFR pathway.

### 3.2. PTX3 Reduces the Angiogenic Potential of MM Cells 

We next assessed the capacity of PTX3 released by MM cells to impair the pro-angiogenic potential of MM in vivo. To this aim, KMS-11 PTX3 cells were grafted onto the top of the chick embryo chorioallantoic membrane (CAM) in the absence or presence of DOXA. As shown in Figure 2A, untreated KMS-11 grafts induced a strong pro-angiogenic response, as shown by the numerous newly formed thin microvessels converging in a spoke-wheel pattern versus the MM cell implant. Notably, this angiogenic response was significantly inhibited by the release of PTX3 from KMS-11 cells following DOXA treatment (Figure 2A). 

To confirm these observations, GFP-expressing MM.1S cells were incubated for 48 h in the absence or presence of DOXA and then grafted into the perivitelline space of zebrafish embryos. As shown in Figure 2B, MM.1S cells induced the formation of EC sprouts originating from the subintestinal vein vessels that were significantly reduced in their length by PTX3 production following DOXA pre-treatment. These data confirm the capacity of MM cells to induce strong pro-angiogenic responses and indicate that PTX3 is able to reduce the angiogenic potential of MM cells in vivo.

### 3.3. MM-Released PTX3 Inhibits Tumor Angiogenesis and Growth In Vivo 

To assess the effect of PTX3 released by MM cells on tumor vascularization and growth, KMS-11 PTX3 cells were grafted subcutaneously in immunodeficient mice. In order to detect the whole functional tumor vascular network, endothelial cells were biotinylated in vivo by i.v. injection of sulfobiotin [25]. As shown in Figure 3A, tumors from mice receiving DOXA in the drinking water showed widespread expression of PTX3 that accumulates in the extracellular matrix. Interestingly, PTX3 strongly reduced FGFR1 activation in both sulfobiotin^+^ endothelial cells and tumor cells as assessed by phospho-FGFR1 immunostaining (Figure 3A). This caused a significant reduction of tumor endothelial cell proliferation as assessed by Ki67/sulfobiotin double immunostatining (Figure 3B). Accordingly, PTX3 expressing xenografts (+DOXA) showed reduced tumor vascularization (Sulfobiotin^+^ area) and proliferation (pHH3^+^area) (Figure 3C) that resulted in a significant delay of tumor growth when compared to controls (−DOXA) (Figure 3D). Of note, DOXA did not affect the growth of mock-transfected KMS-11 tumor grafts, thus ruling out any effect exerted by DOXA per se (data not shown). 

### 3.4. PTX3 Reduces BM Niche Vascularization and Colonization by MM Cells

Since BM angiogenesis plays a pivotal role in MM progression and dissemination [3,4], we investigated the effect of PTX3 on the capacity of MM cells to induce BM angiogenesis while growing in their own microenvironment. To this aim, we took advantage of the MM.1S systemic model of MM by which BM infiltration is detectable 2 weeks after intravenous injection of MM cells, and reaches a peak after 6–8 weeks, when more than 40% of the BM cell population is represented by neoplastic cells [11]. Thus, luciferase-expressing MM.1S cells transduced to express DOXA-inducible PTX3 were systemically injected into SCID beige mice. Notably, the induction of PTX3 expression and release by MM.1S cells (see CD38/PTX3 double immunostaining in Figure 4A) significantly reduced the vascularization of tumor foci growing in the BM, as assessed by CD38/KDR double immunostaining of femurs from mice receiving DOXA in the drinking water (Figure 4A). The reduction of tumor vascularization was confirmed also by 3D two-photon fluorescent imaging of whole femurs (Figure 4B). In keeping with these findings, MM.1S-released PTX3 strongly reduced the systemic spreading and BM colonization of the disease, as detected by in vivo bioluminescence imaging (Figure 4C) and CD38 immunostaining of femur sections (Figure 4D).

## 4. Discussion

BM angiogenesis is a hallmark of MM progression and represents a prognostic factor for MM patients [5,6]. Indeed, like solid tumor cells, highly proliferating plasma cells are able to induce a neovascular response via the release of angiogenic cytokines, giving rise to the so-called “angiogenic switch” [2]. The induction of BM angiogenesis may thus favor the progression from the pre-neoplastic MGUS and non-active MM to active MM, the latter representing the vascular phase of plasma cell tumors. Different mechanisms may take place in order to activate BM angiogenesis and counterbalance the physiological/steady state equilibrium in favor of pro-angiogenic activators. In this context, an analysis performed on BM plasma has revealed an unbalanced ratio between the pro-angiogenic growth factor FGF2 and one of its natural inhibitors, PTX3, in MM patients compared to MGUS patients [21]. This finding suggests that the ratio between PTX3 and FGF2 released from plasma cells and BMSCs decreases during the transition from MGUS to MM, leading to more abundant levels of free FGF2 in the BM. This “stoichiometric” unbalance may actively contribute to pathological BM angiogenesis in MM. 

Based on these observations, we hypothesized that the restoration of a balanced ratio between FGF2 and PTX3 in the BM may be able to restrain the avascular–vascular transition in MM. So far, only in vitro data using recombinant PTX3 have been reported showing that addition of exogenous PTX3 reduces the activation of MM-derived ECs and fibroblasts stimulated by FGF2. 

Here, we demonstrate and confirm the pivotal role played by the PTX3/FGF2 ratio in MM growth, dissemination, and neovascularization exploiting in vivo MM models characterized by the inducible overexpression of PTX3 by human plasma cells. Our data confirm that MM cells are capable to induce, per se, a strong angiogenic response both in vitro and in vivo by activating FGFR1 in ECs. Interestingly, when PTX3 expression is forced in neoplastic plasma cells, thus leading to an increase of the PTX3/FGF2 ratio, the activation of FGF/FGFR pathway is inhibited and EC activation is reduced both in in vitro co-cultures and in in vivo tumor graft models (i.e., chick embryo CAM, zebrafish and murine xenografts). It must be pointed out that GEP analysis performed on PTX3-expressing MM cells indicates that (i) the increased PTX3/FGF2 ratio appears to be due to PTX3 overexpression and not to PTX3-mediated FGF2 downregulation and (ii) the antiangiogenic response is due to the FGF-trap activity exerted by PTX3 rather than to a modulation of the expression of other pro- or anti-angiogenic factors caused by PTX3 overexpression. As a result of this antiangiogenic effect, MM growth and dissemination is significantly impaired in both subcutaneous and systemic murine models using MM cells overexpressing PTX3. 

Beside the antiangiogenic activity exerted by MM-released PTX3, we cannot rule out the possibility that a direct “autocrine” effect exerted by PTX3 overexpression on MM cells may contribute to the observed inhibition of tumor growth and dissemination. Indeed, FGF2 is known to play a pivotal role in the survival and proliferation of MM cells [11]. However, the in vitro data hereby reported, and previous observations from others [21], have shown that endogenous PTX3 overexpression or recombinant PTX3 treatment do not affect the survival and proliferation of MM cells under standard cell suspension culture conditions. On the other hand, we have observed a significant reduction in the expression levels of the proliferation marker *Cyclin D1* when MM cells overexpressing PTX3 were grown embedded in Matrigel (Appendix A). Accordingly, a reduced rate of MM cell proliferation paralleled by a reduced FGFR1 activation occurred in vivo in both endothelial and tumor cells when PTX3 was overexpressed and accumulated in the extracellular matrix of grafted tumors. Together, these data suggest that an interaction with extracellular matrix component(s) may be required to consent and favor the autocrine biological function of PTX3 on MM cells themselves. Further experiments will be required to elucidate this hypothesis.

Altogether the data hereby reported highlight for the first time the role played by endogenous PTX3 released by MM cells on BM-niche components, reinforcing the concept that the PTX3/FGF2 ratio is a key rheostat in MM angiogenesis and progression. In keeping with these findings, we have recently reported a significant reduction of tumor vascularization of subcutaneous KMS-11 xenografts in mice treated with the PTX3-derived small molecule NSC12 [11]. Hence, anti-angiogenic approaches based on PTX3-derived FGF trap molecules may represent a promising future area of research in the field of MM therapy. 

## 5. Conclusions

Our data highlight the role played by endogenous PTX3 released by MM cells on endothelial BM-niche components and demonstrate that the restoration of a balanced ratio between FGF2 and PTX3 affects MM angiogenesis, growth, and progression. Hence, the exploitation of PTX3-derived anti-angiogenic approaches may represent a promising future area of research in the field of MM therapy. 

## Figures and Tables

**Figure 1 cancers-13-02255-f001:**
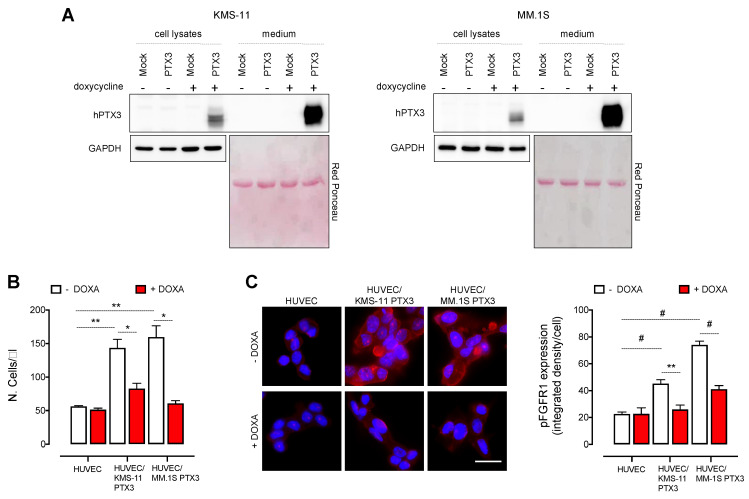
PTX3 released by MM cells impairs HUVEC proliferation by inhibiting FGFR activation. (**A**) Western blot analysis of PTX3 expression (cell lysates) and release (medium) from KMS-11 and MM.1S cells transduced with a doxycycline (DOXA)-inducible PTX3 (PTX3) or a control vector (Mock) and treated or not with DOXA for 48 h. (**B**) Cell count by cytofluorimetric analysis of HUVEC co-cultured or not with KMS-11 PTX3 or MM.1S PTX3 cells for 48 h in the presence or absence of DOXA. (**C**) Left panel: Immunofluorescence analysis of phospho-FGFR1 (red fluorescence) expression in HUVEC co-cultured or not with KMS-11 PTX3 or MM.1S PTX3 cells for 24 h in the presence or absence of DOXA. Scale bar: 50 μm. Right panel: Fluorescence intensity quantification of phospho-FGFR1 by ImageJ software. For each microscopic field, fluorescence intensity values were normalized with the number of nuclei detected by DAPI staining. Data are mean ± SEM of 3 experimental replicates. * *p* < 0.05, ** *p* < 0.01, # *p* < 0.001.

**Figure 2 cancers-13-02255-f002:**
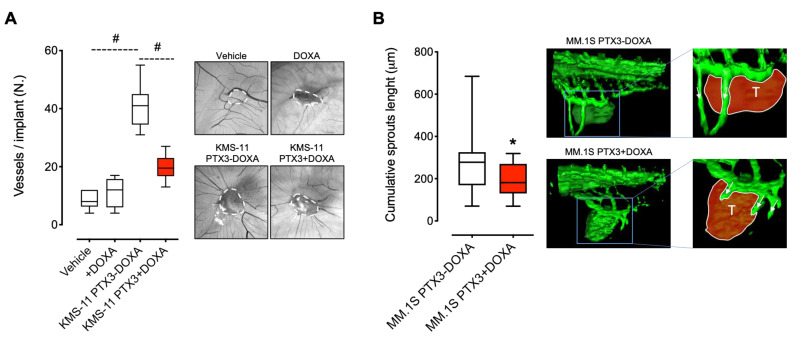
MM cells producing PTX3 are less pro-angiogenic. (**A**) KMS-11 PTX3 cells embedded in alginate pellets containing (KMS-11 PTX3+DOXA) or not (KMS-11 PTX3-DOXA) doxycycline were grafted onto the top of the chick embryo CAM at day 11 of development. PBS (Vehicle) or DOXA were used as control. At day 14, for each embryo, the number of newly formed blood vessels converging towards the implant were quantified. *n* = 15 embryo/group. Representative images of CAMs at day 14 are reported. White dashed lines show the alginate pellet implants. (**B**) GFP-expressing MM.1S PTX3 cells in vitro induced (MM.1S PTX3+DOXA) or not (MM.1S PTX3-DOXA) with DOXA were grafted into the perivitelline space of 48 hpf Tg (fli1:egfp) zebrafish embryos. Twenty-four hours after engraftment, for each embryo, the cumulative length of sprouts deriving from subintestinal vein vessels was quantified. In the magnified images, the tumor mass is highlighted in red and the vessel sprouts are indicated with arrows. *n* = 30 embryo/group. In box and whiskers graphs, boxes extend from the 25th to the 75th percentiles, lines indicate the median values and whiskers indicate the range of values. * *p* < 0.05, # *p* < 0.001.

**Figure 3 cancers-13-02255-f003:**
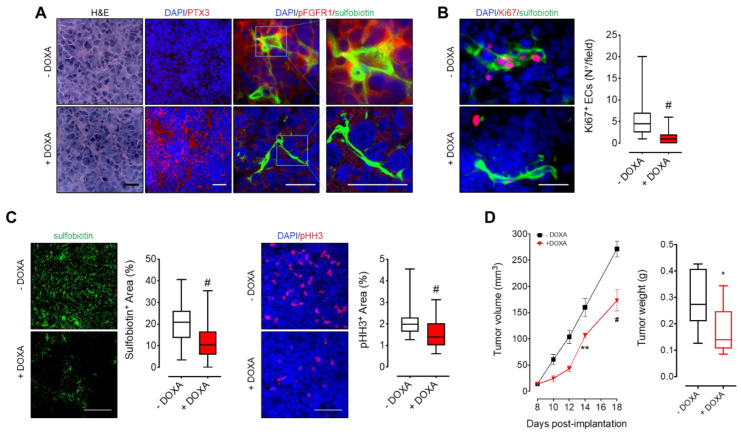
PTX3 released by MM cells reduces tumor vascularization and growth. KMS-11 PTX3 cells were subcutaneously engrafted in NOD/SCID mice receiving (+DOXA) or not (-DOXA) doxycycline in the drinking water. (**A**–**C**) Histological analyses of tumor sections eighteen days after tumor engraftment. Before sacrifice, mice were injected i.v. with sulfobiotin in order to label the whole functional vascular network. Sulfobiotin and phospho-HH3 positive area were quantified by ImageJ software. Scale bar A, B: 50 μm; scale bar C: 100 μm. (**D**) Left panel: Tumor volumes (mean ± SEM) measured with caliper up to 18 days after tumor implantation. *n* = 8 mice/group. Right panel: Tumor weights at day 18 post-implantation. In box and whiskers graphs, boxes extend from the 25th to the 75th percentiles, lines indicate the median values and whiskers indicate the range of values. * *p* < 0.05, ** *p* < 0.01, # *p* < 0.001.

**Figure 4 cancers-13-02255-f004:**
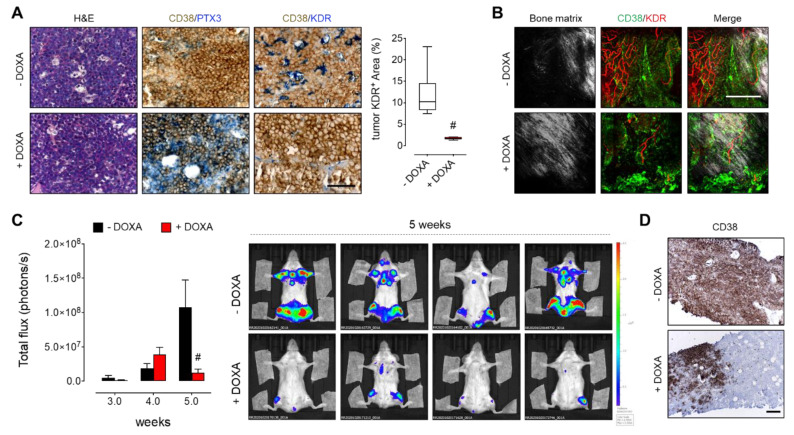
PTX3 reduces MM cell BM colonization. Luciferase-expressing MM.1S PTX3 cells were injected i.v. in SCID Beige mice receiving (+DOXA) or not (-DOXA) doxycycline in the drinking water. (**A**) Histological analysis of femur sections five weeks after tumor cell injection. Tumor cells are detected in brown (CD38) and PTX3 and tumor vessels (KDR) are detected in blue. Scale bar: 100 μm. KDR positive area in tumor spots was quantified by ImageJ software. In box and whiskers graphs, boxes extend from the 25th to the 75th percentiles, lines indicate the median values and whiskers indicate the range of values. # *p* < 0.001. (**B**) Two-photon fluorescence microscopy analysis of femurs five weeks after tumor cell injection. Tumor cells are detected in green (CD38), tumor vessels (KDR) in red, and bone matrix is detected by second harmonic generation in grey. Scale bar: 200 μm. (**C**) Left panel: Quantification of bioluminescent signal of luciferase-expressing MM.1S cells up to 5 weeks after i.v. injection. Data are mean ± SEM, # *p* < 0.001. *n* = 8 mice/group. Right panel: Representative bioluminescence imaging of mice five weeks after MM cell i.v. injection. (**D**) Immunohistochemical analysis of femur sections five weeks after tumor cell injection. Tumor area are detected in brown (CD38). Scale bar: 200 μm.

## Data Availability

Data is contained within the article or Appendix A.

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
