# Peer review of "Pentraxin 3 Inhibits the Angiogenic Potential of Multiple Myeloma Cells"

_cancers, 2021, doi:10.3390/cancers13092255_

Round 1

Reviewer 1 Report

Ronca and colleagues investigated the effects of Pentraxin 3 overexpression in multiple myeloma cell lines on angiogenesis in vitro and in vivo models and the resulting effects on tumor growth in mice. The manuscript is overall well written. As FGF2/Pentraxin ratios are of known importance for multiple myeloma progression, also FGF2 levels should be determined in the study.

Several additional minor points should be considered:

Line 83: Doxycycline concentration is missing.

Line 187: The reference should be in standard format.

Line 216: Doxycycline concentration is described as 200ng/ml, in the methods section it is 200ug/ml. Please clarify.

Line 222: Should read Figure 1 B, C instead of Figure 2.

Line 239: “Together, these data are in keeping with the capacity of FGF2 to act as a paracrine survival/proliferation factor for BM ECs…” is misleading as HUVECs were used.

Figure 2: The labelling is distorted. A: Pictures from vehicle and Dox groups should be provided B: Cumulative sprout length in mm seems to be wrong. The tumor is not indicated in red as described in the Legend.

Figure 3A: Scale bars are missing in the right panel. Figure 3B and C: Why is the Ki67 signal so different from the pHH3 and no tumor cells are positive?

Figure 4: Why the CD38 staining is so different in the +Doxa middle and right panels? CD38 is much less convincing in B compared to A.

Author Response

Response in the attached file.

Reviewer 2 Report

Ronca et al analyzed the anti-angiogenic role of PTX3 produced by MM cells and showed that the inducible expression of PTX3 is able to impair MM neovascularization, the onset of a proficient BM vascular niche and, to impair tumor growth and dissemination.

This is an interesting approach for potential therapeutic translation into the clinic, yet few questions should be addressed.

Major:

In which other especially healthy tissues might Ptx3 be expressed? For therapeutic aspect, this might be of importance (human protein atlas: parts of the brain express Ptx3 or in testis in high levels).

Given that the NSC12 compound is working efficient similar like the knockdown of Ptx3 and could be used as anti-myeloma drug. How would you rate the chance of successful treatment in patients (cell line models) that are resistant against current therapeutic regimen like IMiDs like Lenalidomide or PIs like Bortezomib or Carfilzomib? Would you be still able to address them?

Minor: some fonts in figures have a problem (Fig 2 & 3, Supp Fig 1 &2) with the font.

Author Response

Response in the attached file.

Reviewer 3 Report

The current article entitled," Pentraxin 3 inhibits the angiogenic potential of multiple myeloma cells” is quite interesting and has scientific merits to be considered for publication.         

 Authors shows that FGFs play a pivotal role in the angiogenic switch. Accordingly, the natural FGF antagonist Long Pentraxin 3 is able to reduce the activation of BM stromal components induced by FGFs. They also showed first time, the anti-angiogenic role of PTX3 produced by MM cells demonstrating that the inducible expression of PTX3 is able to impair MM neovascularization, the onset of a proficient BM vascular niche and, ultimately, to impair tumor growth and dissemination. Altogether paper is well written, results support the conclusion.

Having said this, I have few points before publication:

Is there any variation in the BMSCs’ ability to induce EC migration after inducible expression of PTX3?  

FGF2-dependent angiogenesis and inflammation. Since PTX3   Contribution of inflammatory cells in promoting FGF2-dependent angiogenesis

Does PTX3 released by MM cells have any effect on  cell proliferation even after inhibiting FGFR activation?

The BM microenvironment in MM is hypoxic, and hypoxia inducible factor 1α is upregulated in patient MMCs. Is there in alteration of hif when expression of PTX3  induced.

Does Pentraxin-3  affects MM secretory profile? Is it possible to check Expression of both angiogenic and antiangiogenic factors Q-PCR.

Author Response

Response in the attached file.

Round 2

Reviewer 1 Report

The authors responded to the questions . I have no further comments.

Reviewer 3 Report

Please accept the manuscript for the publication. I  have no further  comments. Authors addressed my all concern with references.